# Preparation of Copper Nitride Films with Superior Photocatalytic Activity through Magnetron Sputtering

**DOI:** 10.3390/ma13194325

**Published:** 2020-09-28

**Authors:** Aihua Jiang, Hongjuan Shao, Liwen Zhu, Songshan Ma, Jianrong Xiao

**Affiliations:** 1College of Science, Guilin University of Technology, Guilin 541004, China; jah@glut.edu.cn (A.J.); 2004134@glut.edu.cn (H.S.); zhuliwen1995@163.com (L.Z.); 2School of Physics and Electronics, Central South University, Changsha 410083, China; songshan@csu.edu.cn

**Keywords:** Cu_3_N films, magnetron sputtering, gas flow ratio, photocatalytic activity, methyl orange

## Abstract

TiO_2_ possesses a wide forbidden band of about 3.2 eV, which severely limits its visible light absorption efficiency. In this work, copper nitride (Cu_3_N) films were prepared by magnetron sputtering at different gas flow ratios. The structure of the films was tested by scanning electron microscope, X-ray diffractometer, and X-ray photoelectron spectroscope. Optical properties were investigated by UV-vis spectrophotometer and fluorescence spectrometer. Results show that the Cu_3_N crystal possesses a typical anti-ReO_3_ crystal structure, and the ratio of nitrogen and Cu atoms of the Cu_3_N films was adjusted by changing the gas flow ratio. The Cu_3_N films possess an optical band gap of about 2.0 eV and energy gap of about 2.5 eV and exhibit excellent photocatalytic activity for degrading methyl orange (degradation ratio of 99.5% in 30 min). The photocatalytic activity of Cu_3_N mainly originates from vacancies in the crystal and Cu self-doping. This work provides a route to broaden the forbidden band width of photocatalytic materials and increase their photoresponse range.

## 1. Introduction

The accelerating industrialization has led to rapid increase in pollutant emissions and worsened environmental pollution. Effective methods must be developed to eliminate pollution and protect the environment. To resolve environmental problems, scholars have increasingly focused on semiconductor photocatalysis, which is a safe, simple, and non-secondary green technology. TiO_2_ exhibits excellent photocatalytic properties and has been widely used in various fields in industry and agriculture [1,2,3,4,5]. However, the photocatalytic material of TiO_2_ has a narrow photoresponse range (band gap of approximately 3.2 eV, which can only absorb ultraviolet light) and low photocatalytic quantum efficiency [2,6,7,8,9]. CdS has a small forbidden band width and good matching with the near-ultraviolet light segment in the solar spectrum, but it is prone to photo-corrosion and has a limited service life [9,10,11,12,13]. Therefore, in research of film materials, exploratory work on photocatalytic performance is of considerable importance.

Copper nitride (Cu_3_N) possesses a special anti-ReO_3_ lattice structure, and the central vacancies of the lattice are easily filled by other atoms, leading to changes in the electrical properties and the feasibility of the crystal to be used from the insulator to the conductor [14,15,16,17,18,19,20,21]. At present, research on Cu_3_N films mainly focuses on the influence of preparation parameters on the crystal structure. Changes in the Cu_3_N structure or doping of elements could lead to alterations in electrical and optical properties [16,21,22,23]. The band gap of Cu_3_N films can be adjusted to a relatively large range. Numerous studies have shown that the optical band gap of Cu_3_N films ranges from 1.1 eV to 1.9 eV [15,24,25,26] and varies according to experimental conditions and process parameters. In general, many reports have been published on the regulation of the band gap of Cu_3_N films by doping and changing the nitrogen partial pressure [19,27]. The intrinsic optical band gap of Cu_3_N films is 2.6 ± 0.3 eV [28]. Studies on the application of the Cu_3_N films have mainly focused on the following aspects: Disposable optical storage materials [29,30], barrier layer of low-reluctance tunnel junction [31], field emission materials [32], and lithium-ion battery materials [33,34]. At present, no related literature reports the use of Cu_3_N films as photocatalytic materials.

This work studied the catalytic degradation of methyl orange of Cu_3_N films prepared by magnetron sputtering for the first time and analyzed the catalytic mechanism of the films.

## 2. Experiment

Cu_3_N films were prepared on single-crystal silicon and quartz (size, 2.5 cm × 1.0 cm × 120 nm) substrates by radio frequency (RF) reactive magnetron sputtering (designed and developed by Shenyang Scientific Instrument Co., Ltd., Chinese Academy of Sciences, Shenyang, China) by using a high-purity Cu target (purity of 99.999%) and nitrogen (99.99%) and argon (99.99%) as working gases. The monocrystalline silicon wafer (100) and quartz wafer were washed with acetone and alcohol and rinsed repeatedly with deionized water. The substrate was dried in an oven. The substrate was placed on the turntable at the top of the vacuum chamber, and its speed was 30 rpm. The distance between the target and the substrate was about 18 cm. The distance between the target and the substrate was about 18 cm. A combination of mechanical pump and molecular pump was used to vacuum. At room temperature, the vacuum degree of the vacuum chamber could reach 1.0 × 10^−3^ Pa in about 60 min. The surface of the target was sputter cleaned before deposition. The gas flow ratio *r* (*r* = [N_2_]/[Ar]) was set to 1/1, 1/2, 1/3, and 1/6, and the total gas flow was 40 sccm. The gas was precisely controlled by a flow meter, mixed and sent into the vacuum chamber at room temperature. The RF sputtering power was kept constant at 250 W. The deposition pressure was 1.0 Pa, and the deposition time was about 5.0 min. The films thickness was monitored by the EQ-TM106 (Hefei Kejing Material Technology Co. Ltd., Hefei, China) film thickness monitor (with an accuracy of 0.14 nm), and they were all controlled at a thickness of about 120 nm.

To judge the photocatalytic activity of the Cu_3_N films, we compared their catalytic effect with TiO_2_ films, which were deposited by magnetron sputtering with TiO_2_ (99.99%) as target. The conditions of the TiO_2_ films samples are as follows: The Ar flow was fixed at 40 sccm, and the RF sputtering power, deposition pressure, and deposition time were 250 W, 1.0 Pa, and 5.0 min, respectively. During the TiO_2_ and Cu_3_N film preparation, the substrate was not heated.

The surface morphology of the Cu_3_N films was observed by scanning electron microscopy (SEM, JSF-2100, HITACHI, Tokyo, Japany), and the surface energy disperse spectra (EDS) of the films were recorded. The phase composition of the films on the silicon substrate was tested by X-ray diffractometer (XRD, X’pert3 Powder, PANalytical, Lelyweg, the Netherland) using Cu–Kα radiation (λ = 0.154 nm), and the scanning angle range was 20°–70° at a 0.015° step size. Surface photoelectron spectroscopy of the films was carried out by X-ray photoelectron spectroscope (XPS, ESCALAB 250Xi model, Boston, MA, USA) with a monochromatic Mg-K X-ray source, and the spectra were calibrated by carbon peak C 1s at 284.5 eV. Due to the limitation of the experimental conditions, the latest reported method was not used for calibration [35]. The transmission spectrum of the films was recorded using UV-vis spectrophotometer (UV-2700, Shimadzu, Kyoto, Japan) in the wavelength range 300–1200 nm at room temperature. The optical band gap of the films was obtained in accordance with the transmission spectrum of the films combined with extrapolation of the Tauc equation. The photoluminescence (PL) properties of the Cu_3_N films on silicon wafers were tested using a Cary Eclipse (Agilent Technologies Inc., Santa Clara, CA, USA) fluorescence spectrophotometer with a wavelength of 370 nm He–Cd.

In this work, 50 mL of 20 mg/L methyl orange solution was used as the target degradation product and Cu_3_N films with size of 2.5 cm × 1.0 cm × 120 nm were used as the catalyst. A high-pressure mercury lamp (500 W) was used as light source (the films were 25 cm away from the light source). The effect of the films on the photocatalytic degradation of methyl orange was studied. Sampling and testing were conducted once every 3 min. The absorbance of the methyl orange solution degraded by the films was recorded using UV-vis spectrophotometer (UV-2700, Shimadzu, Kyoto, Japan).

## 3. Results and Discussion

Figure 1a–d shows the surface morphology and cross-section of the SEM image of Cu_3_N films prepared at different values of *r*. The films prepared have flat and compact surface. However, as *r* increases, the number of large particles on the films surface increases significantly. The reason may be that as *r* increases, the content of Cu_3_N in the films decreases, and the elemental Cu increases accordingly. When *r* is low, part of the sputtered Cu atoms have no time to react with the nitrogen atoms and are deposited on the substrate. Therefore, Cu_3_N films are mainly formed by forming Cu-N bond by absorbing the nitrogen atoms and inserting a lattice of the Cu atoms. At this time, a considerable part of the Cu atoms on the surface of the films does not bond to nitrogen. Therefore, the central vacancy of the Cu_3_N lattice still contains more Cu atoms. The lattice constant of the Cu_3_N crystal is smaller than that of Cu. Many studies have reported the occurrence of preferential growth in the preparation of Cu_3_N films by magnetron sputtering [15,18,27,28,32]. This preferential growth phenomenon indicates the strong preferential growth mechanism of crystal grains during the growth of the Cu_3_N films. In addition, the diffusion process causes lattice mismatch stress, resulting in a rough surface morphology. Only when the preferential and diffusion mechanisms during growth reach the equilibrium, the films’ surface tends to be flat. The cross-section image (Figure 1e) shows that the Cu_3_N films are not dense and have some pore defects.

Figure 2a shows the surface energy disperse spectra (EDS) of Cu_3_N films prepared at *r* = 1/3. The atomic percentages of various elements in the films can be obtained (see the insert table in figure). The surface of the films contains low amount of oxygen and carbon contaminations. We believe that it mainly originated from the adsorption of the films surface before the test. In the case of *r* = 1/3, the Cu and nitrogen ratio in the films is approximately 3.85:1. The copper–nitrogen atomic ratio value is larger than the copper–carbon ratio of 3:1 in the chemical formula of Cu_3_N, indicating the presence of Cu-rich atoms in the films. The reasons may be as follows: First, some Cu atoms combine with oxygen atoms in the vacuum chamber to form Cu-O, and oxygen originates from the background atmosphere of the vacuum chamber. Second, a very small amount of Cu atoms is deposited directly on the substrate before it reacts with nitrogen in the vacuum chamber. In addition, other phases such as Cu_4_N may exist in the films [36], causing it to deviate from the stoichiometric ratio of 3:1. The corresponding EDS elemental mapping of the surface of the Cu_3_N films is shown in Figure 2b.

Figure 3a shows the XRD results of the Cu_3_N films. The diffraction peaks of the crystal planes of Cu_3_N crystals are sharp, which indicates that the Cu_3_N films prepared by magnetron sputtering has a good crystallinity under different *r*. When *r* is small, the Cu_3_N films show preferential growth in the crystal planes of (100) and (200). By contrast, the films show preferential growth the crystal plane of (110). This result is similar to most research reports, which indicate that Cu_3_N prepared by magnetron sputtering has an anti-ReO_3_ structure [19,31,37]. The reason for the preferential growth orientation of the Cu_3_N films is as follows: At low *r*, films are mainly formed by Cu-N bonds due to the adsorption of nitrogen atoms into the crystal lattice of Cu atoms; and then Cu_3_N (100) and Cu_3_N (200) planes are grown. At high *r*, sufficient nitrogen atoms combine with Cu on the target surface or substrate surface to form Cu-N. The Cu_3_N (110) plane is grown on the films according to the principle that the free energy of the crystal plane has the lowest priority. Another explanation is that as the flow of nitrogen increases, the number of Cu atoms reaching the surface of the substrate decreases. As such, insufficient Cu atoms combine with high-energy nitrogen, resulting in the weakening of the Cu_3_N (110) peak intensity [28,31]. When the experimental parameters change, the preferential growth of the Cu_3_N crystal plane shifts from (100) to (111) [38]. This phenomenon may be caused by different preparation methods employed. At the same time, it is not difficult to find that there are two very weak diffraction peaks around 31.5° and 48.1°, which belong to the diffraction peaks of CuO and Cu, respectively [39]. The XRD diffraction pattern of the TiO_2_ films is shown in Figure 3b. The peaks at 25.3°, 38.0°, 48.1°, and 53.3° correspond to the diffraction peaks of the (101), (112), (200), and (211) crystal planes of TiO_2_ [8,40], respectively, which belong to anatase TiO_2_ as referred to JCPDS card (NO. 21-1272). The peaks at 54.8° correspond to the diffraction peak of the (211) crystal plane of TiO_2_, belonging to the tetragonal rutile phase [9]. The main body of the TiO_2_ films prepared by magnetron sputtering is anatase structure. Compared with rutile, anatase has more oxygen vacancies and can easily trap electrons, resulting in better electron–hole separation effect [41].

Figure 4a shows the survey scan of the XPS spectrum of Cu_3_N films prepared at different *r* values. A strong oxygen peak appears in the spectrum, indicating that the surface of the films contains more oxygen. Oxygen mainly originates from the moisture and atmospheric absorption of the films and the residual oxygen in the vacuum chamber. As the *r* increases, the CuLMM peak and Cu2P peak in the films increase significantly, thereby indicating that increasing flow rate of N_2_ is more favorable for the formation of a Cu_3_N structure by nitrogen and Cu atoms in the vacuum chamber. To analyze the bonding mode and valence state of Cu in the films, we performed Lorentz fitting on the Cu2P peak of the XPS spectrum. Figure 4b shows the XPS fitting of Cu2P for the Cu_3_N films. In the range of 930.0–940.0 eV, the spectral line can be fitted by two Cu characteristic peaks, namely, Cu^2+^ Cu2P_3/2_ (334.5 eV) and Cu^1+^ Cu2P_3/2_ (333.0 eV), indicating that the valence of Cu on the surface of the films is +2 or +1 [42,43]. At the same time, according to the fitting results, we obtained by simple estimation that the ratio of Cu^1+^ to Cu^2+^ in the films deposited under the conditions of *r* = 1/1 and *r* = 1/3 is 3.66 and 0.69, respectively. As the nitrogen pressure in the vacuum chamber decreases, the Cu^+^ component in the films decreases. Figure 4c shows the N1s in the Cu_3_N films. The peak of binding energy of N is mainly at 400.0 eV, and the peak at 397.8 eV is relatively small. The binding energy at 397.8 eV indicates that the nitrogen atom in the form of beta-N (*β*-N) exists in the crystal, and that at 400.0 eV indicates that the nitrogen atom in the form of gamma-N (*γ*-N) exists in the crystal or in the form of N-N and N-O bond [44]. Similarly, according to the Lorentz fitting results of the N1s peak, the ratio of *β*-N to *γ*-N in the films deposited under the conditions of *r* = 1/1 and *r* = 1/3 is estimated to be 2.68 and 2.03, respectively.

Figure 5a shows the UV-vis absorption spectrum of the Cu_3_N films. The films show good absorption in the visible light band, indicating its feasibility for catalytic application to visible light. The absorption coefficient α of the Cu_3_N films can be obtained directly by the absorption spectrum of the films as follows:(1)$$\alpha =\ln\left(\frac{100}{T}\right)/d$$
where *T* and *d* are the transmittance and thickness of the Cu_3_N films. The optical band gap (*E_g_*) of the Cu_3_N films can be obtained by the Tauc equation as follows:(2)$$\alpha h\upsilon =A{\left(h\upsilon -E_{g}\right)}^{2}$$
where *h*, *υ*, and *A* are the Planck constant, optical frequency, and constant, respectively. The *E_g_* of the films prepared by magnetron sputtering is within 1.96–2.09 eV (Figure 5b), and the result is slightly larger than that in literature [28,45,46]. The reason may be caused by different preparation techniques and preparation parameters.

The photoluminescence (PL) spectra of Cu_3_N films prepared under different *r* at 370 nm excitation wavelength are shown in Figure 6. The room-temperature PL bands of the Cu_3_N films prepared under different *r* are mainly concentrated in the blue-violet light region. The PL spectrum of the sample shows three peaks that are located at 418, 488, and 507 nm. The peak position at 418 nm is very strong, and the peak positions at 488 and 507 nm are relatively weak. We think that the peak position at 418 nm is a band-edge emission of the Cu_3_N films, and the two other peaks are caused by defect-state fluorescence [28]. The change in *r* has little effect on the intensity of each luminous peak. Based on this spectrum, the intrinsic luminescence energy gap of the Cu_3_N film is within 2.34–2.97 eV, which is higher than the optical band gap *E_g_* obtained in Figure 5b (1.96–2.09 eV) [47,48]. The energy level difference is about 0.6 eV. The light emitting region of Cu_3_N in the blue-violet light range is mainly due to the following: Unbonded Cu atoms (which can be regarded as self-doping) that form the electronic transition from the defect level to the valence band [49]; the energy levels of the center vacancy defects in the Cu nitride crystal and the electronic transitions between the composite defects; and the electronic transitions between the interface defects near the Cu_3_N grain boundary and the valence band [28].

The degradation rate reaches 99.8% when *r* = 1/3. As shown in Figure 7, the photodegradation rate of methyl orange increases with the extension of the illumination time. The photocatalytic degradation rate of methyl orange on the films reaches 93.5% after 30 min of illumination, and *r* = 1/3 is the best photocatalytic activity (degradation rate reaches 99.5%). This finding is due to the large number of voids and free Cu coexisting in the Cu nitride structure prepared at *r* = 1/3, and the combination of the two exhibits good photocatalytic activity. The superior photocatalytic properties of the Cu_3_N films are mainly due to its special anti-ReO_3_ lattice structure. In Cu_3_N crystals, Cu atoms do not occupy the tight position of the lattice (111) plane but leave many voids in their crystal structure, causing the other atoms to easily fill the central vacancy of the Cu_3_N lattice. After this void is filled with other atoms, structural defects or impurity replacement defects are formed in the Cu_3_N crystal. The presence of these defects plays an important role in photocatalysis.

A schematic of the degradation of methyl orange by the Cu_3_N films is shown in Figure 8. In photocatalysis, the formation of •OH free radicals is very important [50]. Each Cu_3_N particle in the films can be regarded as a small short-circuit photoelectrochemical cell, which can generate wide-ranging electrons (*e*^−^) and holes (*h*^+^) under the photoelectric effect. The *e*^−^ and *h*^+^ migrate to different positions on the surface of the Cu_3_N grains under the action of the electric field. The photo-generated electrons *e*^−^ on the surface of the Cu_3_N grains are easily captured by oxidizing substances, such as dissolved oxygen in the water. The *h*^+^ can oxidize organic substances adsorbed on the surface of the Cu_3_N grains or first oxidize ·OH and H_2_O molecules adsorbed on the surface of the Cu_3_N grains into ·OH radicals. The oxidizing ability of ·OH radical is the strongest among the oxidants in the water body, and it can oxidize most of organic matter and inorganic pollutants in the water and mineralize them into inorganic small molecules, carbon dioxide, water, and other harmless substances. The possible chemical formulas are as follows:(3)$${\mathrm{Cu}}_{3}N+hv\to h^{+}+e^{-}$$
(4)$$h^{+}+{\mathrm{OH}}^{-}\to \cdot \mathrm{OH}$$
(5)$$h^{+}+H_{2}O\to \cdot \mathrm{OH}+H^{+}$$
(6)$$e^{-}+O_{2}\to \cdot O_{2}^{-}$$
(7)$$\mathrm{OH}+H^{+}dye\to \cdots \to {\mathrm{CO}}_{2}+H_{2}O.$$

The superior photocatalytic degradation activity of the Cu_3_N films can be explained by frontier molecular orbital theory [51,52]. In this photocatalytic reaction system, the 2P electron orbital of oxygen atom is the frontier molecular orbit (HOMO), and the metal Cu orbital in the films is the lowest unoccupied molecular orbital (LUMO). The electrons are transitioned from the oxygen atom under specific wavelength illumination. For the Cu atom, the atomic HOMO needs to reach stable electrons to form ·OH with e in water molecules, thereby effectively degrading the methyl orange molecule and achieving the photocatalytic effect.

## 4. Conclusions

Cu_3_N films were prepared by magnetron sputtering on silicon and quartz substrate under different gas flow ratios. Characterization and analysis of the microstructure and optical properties of the films were performed using modern analytical testing techniques and methods. The Cu_3_N films prepared in this work have a good crystalline phase and present an anti-ReO_3_ crystal structure. The Cu_3_N films contains Cu-rich atoms and have part of structure vacancies. These defects play a decisive role with regard to the structure and light absorption properties of the films. Experimental results show that under the irradiation of mercury lamp, the degradation rate of methyl orange by the Cu_3_N films can reach more than 99.5% (when *r* = 1/3, the best effect is achieved), which is superior to that of TiO_2_ films with the same thickness prepared by magnetron sputtering. This finding is due to the presence of more defects and Cu-rich atoms in the films, resulting in effective separation of photogenerated electron–hole pairs. This phenomenon also expands the absorption band edge red shift, enhances the light absorption ability, and confers excellent photocatalytic performance. Hence, the fabricated Cu_3_N films can oxidize pollutants into carbon dioxide and water for effective degradation and would be a promising low cost and nontoxic material for photocatalysis.

## Figures and Tables

**Figure 1 materials-13-04325-f001:**
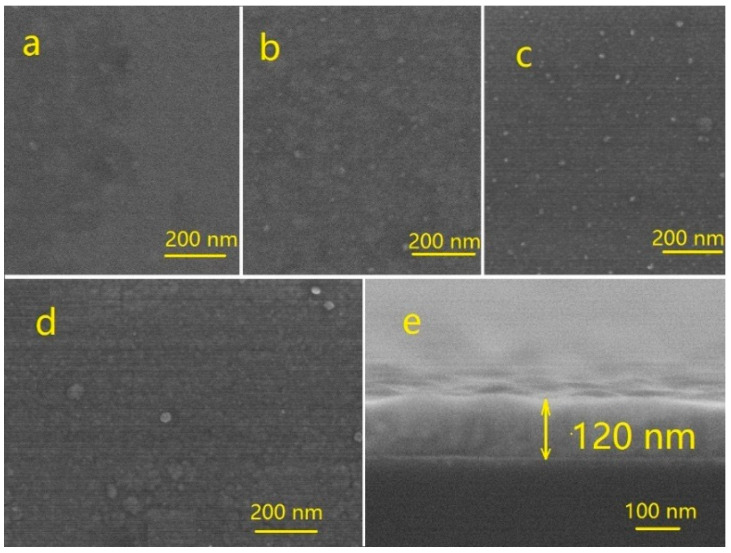
SEM surface morphology of Cu_3_N films prepared at different: (**a**) *r* = 1/6; (**b**) *r* = 1/3, (**c**) *r* = 1/2; (**d**) *r* = 1/1; (**e**) cross-sectional morphology of films prepared at *r* = 1/3.

**Figure 2 materials-13-04325-f002:**
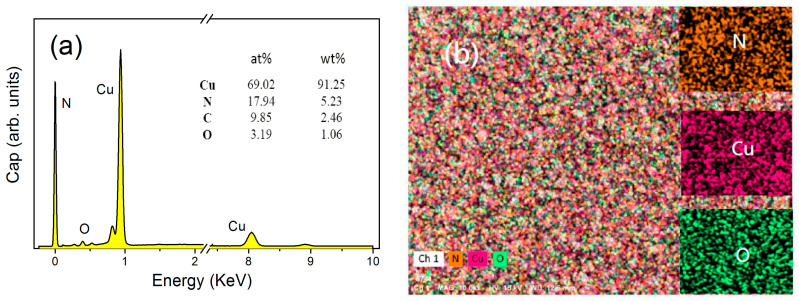
(**a**) Energy disperse spectra (EDS) results of Cu_3_N films prepared at *r* = 1/3, and the inserted value are the atomic and mass percentages of various atoms in the Cu_3_N films; (**b**) corresponding EDS elemental mapping of the surface of the Cu_3_N films.

**Figure 3 materials-13-04325-f003:**
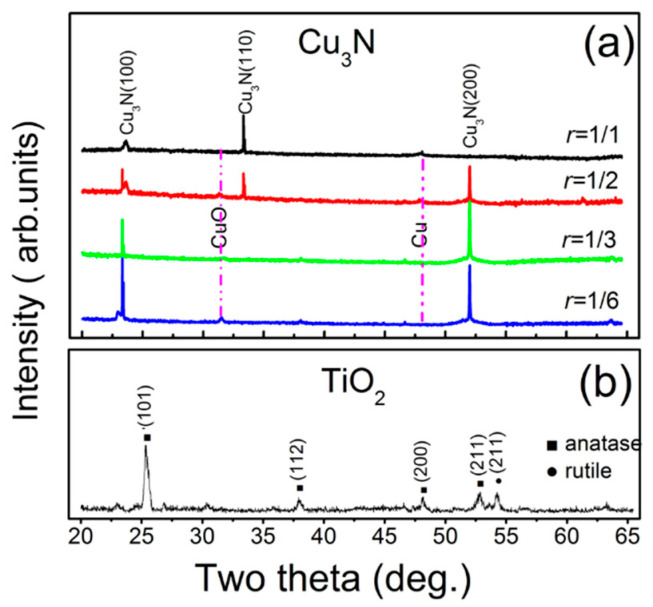
XRD diffraction patterns of films deposited by magnetron sputtering (**a**) Cu_3_N and (**b**) TiO_2_.

**Figure 4 materials-13-04325-f004:**
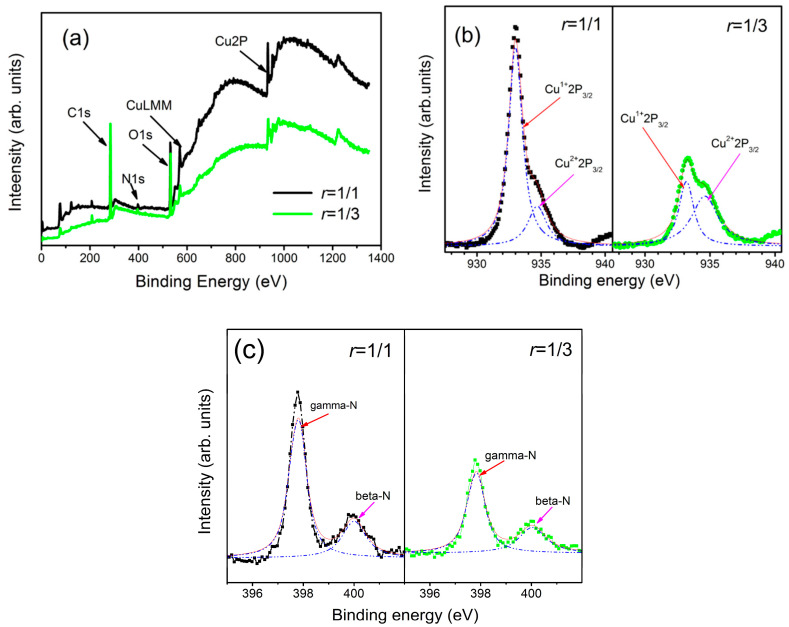
XPS spectra of Cu_3_N films deposited by *r* = 1/1 and *r* = 1/3: (**a**) Survey, (**b**) deconvoluted Cu2p spectra, and (**c**) deconvoluted N1s spectra.

**Figure 5 materials-13-04325-f005:**
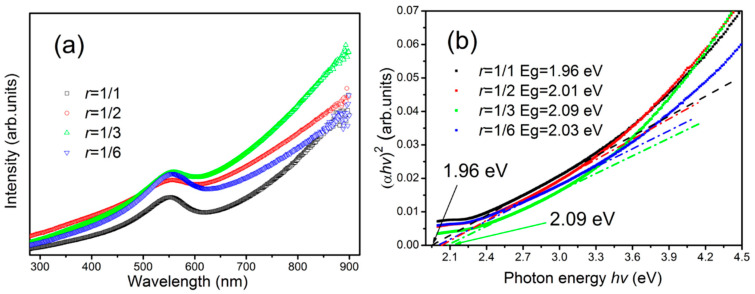
UV-vis transmission spectra of the Cu_3_N films prepared at different *r* (**a**), and determination of the optical band gap of the films (**b**).

**Figure 6 materials-13-04325-f006:**
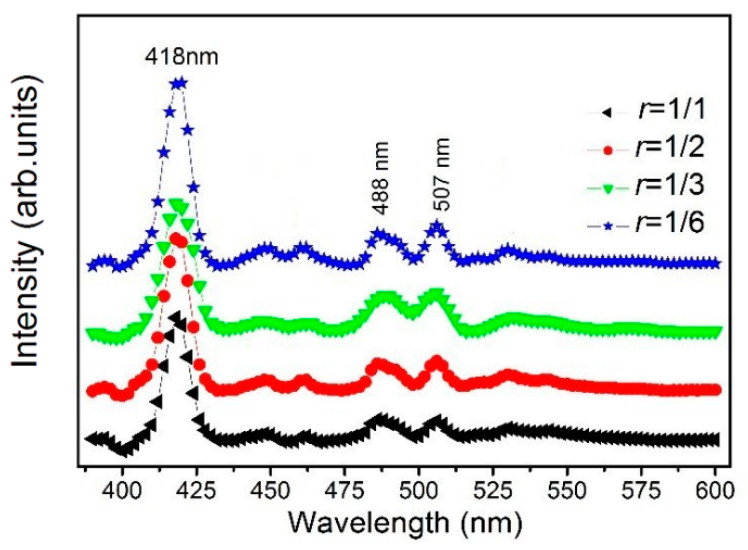
Photoluminescence spectra of Cu_3_N films prepared at different values of *r*.

**Figure 7 materials-13-04325-f007:**
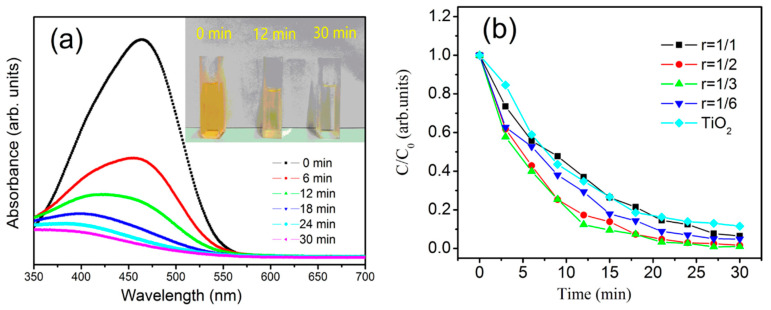
(**a**) Methyl orange absorbance of the films prepared at *r* = 1/3 at different times; (**b**) curve of methyl orange degradation by films prepared at different levels of *r*. For comparison, the degradation curves of the TiO_2_ films are presented.

**Figure 8 materials-13-04325-f008:**
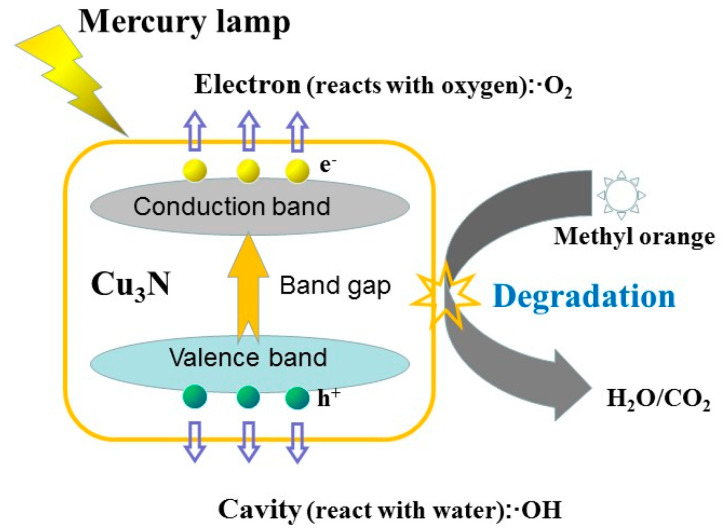
Scheme of photocatalytic degradation of methyl orange solution by the Cu_3_N films.

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
