# Peer review of "Preparation of Copper Nitride Films with Superior Photocatalytic Activity through Magnetron Sputtering"

_materials, 2020, doi:10.3390/ma13194325_

Round 1

Reviewer 1 Report

The manuscript is well written, clear and easy to understand. There are some points in the manuscript that should be addressed.

Fig.1. I suggest to use the AFM surface morphology instead of SEM as it can be more informative for this research.

Fig. 2b should be improved as it is hard to read some text on the image.

Optical band gap calculated only for two Cu3N films (Fig. 5b.). It should be calculated for all prepared Cu3N films.

Author Response

Response to Reviewer comments:

Response to Reviewer #1:

The manuscript is well written, clear and easy to understand. There are some points in the manuscript that should be addressed.

We kindly thank the Reviewer for the evaluation of our work and providing useful comments.

Comment 1. Fig.1. I suggest to use the AFM surface morphology instead of SEM as it can be more informative for this research.

Reply: The suggestions made by the reviewers are quite good. However, due to experimental conditions, samples need to be prepared immediately and sent to institutions for testing. Therefore, it is difficult for us to obtain AFM test results in a short period of time.

Comment 2. Fig. 2b should be improved as it is hard to read some text on the image.

Reply: According to the reviewer's suggestion, Figure 2b has been improved, and it is as follows.

Comment 3. Optical band gap calculated only for two Cu3N films (Fig. 5b.). It should be calculated for all prepared Cu3N films.

Reply: Based on the opinions of reviewers, we calculated the optical band gaps of the films under different r and listed them in Figure 5b. Figure 5b is as follows:

Reviewer 2 Report

The paper reports on the study of Cu3N thin films produced by magnetron sputtering for photocatalytic applications.
The topic is intersting and worth to investigate even if the paper does not report any impressive results.
The manuscript will be suitable for publication in Materials journal after significat revision along the lines listed below.

The orientation of single-crystal silicon substrates in Section 2 should be mentioned as well as
a purity of argon and nitrogen gases used. The thickness of all films should be reported.

The use of self-made TiO2 films as a reference for photocatalytic activity tests is not convincing,
since such films could be a subject of independent study by themselves.
Some standard materials as, for example, Degussa P25 would be much better, at least, concerning with possibility to reproduce.

The details of all experimental setups should be added in Section 2.
For example, the X-ray tube, type of detector and the use of filters & monochromator should be given for XRD,
the source and detectors for XPS.
The model of UV-vis spectrophotometer should be provided the photocatalytic measurements.

It is not clear what the authors mean under "the number of films particles" in line 89?
The mechanism of film growth is not fully clear, especially, concerning the formation of pure metallic and oxide phases.

The phrase "The diffraction peaks of the crystal planes of the Cu3N crystals are sharp and have no peaks."
is very confusing. The origin of a number of small but visible peaks on Cu3N diffraction patterns in Fig. 3(a) should be explained.

The results of XRD and XPS look controversial regarding the presence of Cu2+ ions, probably related to oxide.

The results on the band gap from optical measurements are not convincing due to the quality of experimental data.
The fits reported in Fig. 5(b) are too optimistic. The choice of the fit range is not explained.
Ellipsometric measurements should be considered.

The role of metal and oxides phases in photocatalytic process are not considered.

The conclusion "The Cu3N films contains Cu-rich atoms and have part of structure vacancies. These defects play
a decisive role with regard to the structure and light absorption properties of the films." is not clear and
not suported by the results. The role of oxide is ignored.

The authors are advised to consider the following recent works on Cu3N:
on the structure of magnetron sputtered thin films (DOI: 10.1515/lpts-2016-0011) and bulk Cu3N (DOI: 10.1016/j.actamat.2017.02.074).

Typos should be corrected in the text, e.g., in line 49 "tCu3N".

Author Response

Response to Reviewer comments:

Response to Reviewer #2:

The paper reports on the study of Cu3N thin films produced by magnetron sputtering for photocatalytic applications. The topic is interesting and worth to investigate even if the paper does not report any impressive results. The manuscript will be suitable for publication in Materials journal after significate revision along the lines listed below.

 We thank the Reviewer for the careful consideration of our work and the provided useful comments.

Comment 1. The orientation of single-crystal silicon substrates in Section 2 should be mentioned as well as a purity of argon and nitrogen gases used. The thickness of all films should be reported.

Reply: We thank the Reviewer for the suggestion. As Reviewer suggested that the orientation of single-crystal silicon substrates and the purity of argon and nitrogen gases are added to the manuscript.

“… using a high-purity Cu target (purity of 99.999%) and nitrogen (99.99%) and argon (99.99%) as working gases. The monocrystalline silicon wafer (100) and …”

The films thickness is monitored by the EQ-TM106 film thickness monitor (with an accuracy of 0.14 nm), and they are all controlled at a thickness of about 120 nm.

Comment 2. The use of self-made TiO2 films as a reference for photocatalytic activity tests is not convincing, since such films could be a subject of independent study by themselves. Some standard materials as, for example, Degussa P25 would be much better, at least, concerning with possibility to reproduce.

Reply: The reviewer’s comments are correct, and we very much agree. It is not convincing to use a homemade TiO2 films as a reference for photocatalytic activity testing, and it is definitely better to use standard materials (such as Degussa P25). Considering that the purpose of this article is to study the photocatalytic activity of copper nitride films prepared by magnetron sputtering, our idea is to use the same preparation method. If the TiO2 films prepared by other methods are used, they are worried that they are not comparable.

Comment 3. The details of all experimental setups should be added in Section 2. For example, the X-ray tube, type of detector and the use of filters & monochromator should be given for XRD, the source and detectors for XPS. The model of UV-vis spectrophotometer should be provided the photocatalytic measurements.

Reply: Considering the Reviewer’s suggestion, the details of experimental setups should be added in Section 2.

The phase composition of the films on the silicon substrate was tested by X-ray diffractometer (XRD, X’pert3 Powder) using Cu–Kα radiation (λ = 0.154 nm), and the scanning angle range was 20°–70° at a 0.015° step size. Surface photoelectron spectroscopy of the films was carried out by X-ray photoelectron spectroscope (XPS, ESCALAB250Xi) with a monochromatic Mg-K x-ray source, and the spectra were calibrated by carbon peak C 1s at 284.5 eV. Due to the limitation of the experimental conditions, the latest reported method was not used for calibration [34]. The transmission spectrum of the films was recorded using UV-vis spectrophotometer (UV-2700) in the wavelength range 300–1200 nm at room temperature. The optical band gap of the films was obtained in accordance with the transmission spectrum of the films combined with extrapolation of the Tauc equation. The photoluminescence (PL) properties of the Cu3N films on silicon wafers were tested using a Cary Eclipse fluorescence spectrophotometer with a wavelength of 370 nm He–Cd. The absorption of the photocatalytic degradation of methyl orange was recorded using UV-vis (UV-2700) spectrophotometer.

Comment 4. It is not clear what the authors mean under "the number of films particles" in line 89? The mechanism of film growth is not fully clear, especially, concerning the formation of pure metallic and oxide phases.

Reply: We are very sorry for our incorrect writing, and we modify it as follows:

However, as r increases, the number of large particles on the films surface increases significantly.

Exploring the growth mechanism of the films is indeed very meaningful, we have not fully understood it yet. In the future, we will carry out some research in this area based on the opinions of reviewers.

Comment 5. The phrase "The diffraction peaks of the crystal planes of the Cu3N crystals are sharp and have no peaks. "is very confusing. The origin of a number of small but visible peaks on Cu3N diffraction patterns in Fig. 3(a) should be explained.

Reply: We are very sorry for our incorrect writing, and we modify it as follows:

The diffraction peaks of the crystal planes of Cu3N crystals are sharp, which indicates that the Cu3N films prepared by magnetron sputtering has a good crystallinity under different r.

According to the recommendations of reviewers, we re-identified the diffraction peaks in Figure 3(a). At the same time, the following expression was added to the manuscript:

At the same time, it is not difficult to find that there are two very weak diffraction peaks around 31.5° and 48.1°, which belong to the diffraction peaks of CuO and Cu, respectively [40].

[40] M. Akira,T. Takahiro,K. Nobuhiro, Synthesis of Cu3N from CuO and NaNH2. Journal of Asian Ceramic Societies. 2014, 2(4): 326-328.

Comment 6. The results of XRD and XPS look controversial regarding the presence of Cu2+ ions, probably related to oxide.

Reply: This is a pretty good question, and it really confuses us. Unfortunately, it is difficult for us to answer at this stage. In the XRD pattern, we marked the CuO peak according to the reference, but failed to conduct in-depth analysis and interpretation.

Comment 7. The results on the band gap from optical measurements are not convincing due to the quality of experimental data. The fits reported in Fig. 5(b) are too optimistic. The choice of the fit range is not explained. Ellipsometric measurements should be considered.

Reply: Just as the reviewer experts point out, there are certain random factors in the calculation of the optical band gap of the film using the Tauc equation. However, we now have no other better way.

Unfortunately, we do not currently have the conditions for Ellipsometric testing. In future work, we will consider this good suggestion from reviewers.

Comment 8. The role of metal and oxides phases in photocatalytic process are not considered. The conclusion "The Cu3N films contains Cu-rich atoms and have part of structure vacancies. These defects play a decisive role with regard to the structure and light absorption properties of the films." is not clear and not supported by the results. The role of oxide is ignored.

Reply: As in comment 6, this is exactly where this article needs to be improved, but due to current reasons, we cannot complete it for the time being. Thanks again to the reviewers for their good suggestions.

Comment 9. The authors are advised to consider the following recent works on Cu3N: on the structure of magnetron sputtered thin films (DOI: 10.1515/lpts-2016-0011) and bulk Cu3N (DOI: 10.1016/j.actamat.2017.02.074).

Reply: According to the suggestions of reviewers, the corresponding literature has been cited in the manuscript.

[21] A. Kuzmin, A. Kalinko, A. Anspoks, J. Timoshenko, R. Kalendarev, Study of Copper Nitride Thin Film Structure. Latvian Journal of Physics and Technical Sciences. 2016, 53(2): 31-37.

[22] T. Janis, A. Andris, K. Alexandr, A. Kuzmin, Thermal disorder and correlation effects in anti-perovskite-type copper nitride. Acta Materialia. 2017, 129:61-71.

Comment 11. Typos should be corrected in the text, e.g., in line 49 "tCu3N".

Reply: We are very sorry for our negligence, and typos has been corrected in the text. At the same time, we checked the manuscript carefully.

Reviewer 3 Report

Title of the article is suitable to the presented topic.

Introduction provides sufficient background and include all relevant references. References are underlined - why?

Experiment is presented, but there is no information about the distance of the sample according to the sputtering target. Distance of the target is important, because of the growth arte of the films. Reader does not get the information, how long was the stabilization time when achieving the appropriate pressure --> residual gases and humidity. It is also important to get more detail information on gas mixture input (temperature of mixing chamber). Was the gas put inside the sputtering chamber during the pressure, were the vacuum pumps on and which vacuum pump was working. When presenting sputtering power it is also important to have information about the I and U - it depends on the gas mixture. I suggest to present an experiment with a Figure, for better understanding of experiment.

Results and discussion is presented.

Figure 1 - Readers do not get the impression on how the mixture effects the cross section of the film. Is the film always of the same thickness?

Figure 4 is well presented and understood.

Equations are presented and understood, but not marked.

It is not clear, what is the distance between the sample and the mercury lamp. Does the temperature effect the degradation?

Conclusion is well presented, but can be improved with the answers to the questions in the review.

Author Response

Response to Reviewer comments:

Response to Reviewer #3:

We kindly thank the Reviewer for the evaluation of our work and providing useful comments.

Comment 1. Introduction provides sufficient background and include all relevant references. References are underlined - why?

Reply: The underlined references are automatically generated by the EndNote document database and the underline has no other meaning.

Comment 2. Experiment is presented, but there is no information about the distance of the sample according to the sputtering target. Distance of the target is important, because of the growth rate of the films. Reader does not get the information, how long was the stabilization time when achieving the appropriate pressure --> residual gases and humidity. It is also important to get more detail information on gas mixture input (temperature of mixing chamber). Was the gas put inside the sputtering chamber during the pressure, were the vacuum pumps on and which vacuum pump was working. When presenting sputtering power it is also important to have information about the I and U - it depends on the gas mixture. I suggest to present an experiment with a Figure, for better understanding of experiment.

Reply: Considering the Reviewer’s suggestion, the following content was added to the manuscript experiment section.

The substrate was placed on the turntable at the top of the vacuum chamber, and its speed was 30 rpm. The distance between the target and the substrate was about 18 cm. The distance between the target and the substrate was about 18 cm. A combination of mechanical pump and molecular pump was used to vacuum. At room temperature, the vacuum degree of the vacuum chamber could reach 1.0×10-3 Pa in about 60 min.

The gas was precisely controlled by a flow meter, mixed and sent into the vacuum chamber at room temperature.

In all processes of vacuuming and films deposition, both mechanical pumps and molecular pumps are used.

After the RF power is determined, the current and voltage of the RF power supply we use are automatically matched, and there is no specific current and voltage value.

Comment 3. Results and discussion is presented. Figure 1 - Readers do not get the impression on how the mixture effects the cross section of the film. Is the film always of the same thickness?

Reply: The thickness of the films prepared in this work is basically the same. Related statements are added in the manuscript.

The films thickness is monitored by the EQ-TM106 film thickness monitor (with an accuracy of 0.14 nm), and they are all controlled at a thickness of about 120 nm.

Comment 4. Figure 4 is well presented and understood. Equations are presented and understood, but not marked.

Reply: Considering the Reviewer’s suggestion, equations are marked.

Comment 5. It is not clear, what is the distance between the sample and the mercury lamp. Does the temperature effect the degradation?

Reply: The distance between the sample and the mercury lamp is 25 cm. See line 217 of the manuscript: “the films were 25 cm away from the light source”. In view of the fact that this description appears in the experimental part, the reader's attention can be more noticed. So in the revised version, we changed these to the experimental part.

Comment 6. Conclusion is well presented, but can be improved with the answers to the questions in the review.

Reply: The manuscript was revised with our best efforts based on the suggestions and requirements of reviewers.

Round 2

Reviewer 2 Report

The revision is only partially satisfactory. Several important points remain unclear
and require further studies as authors mentioned in their reply.
Therefore, I cannot recommend the paper for publication in the present form.

There are three main problems:

1) the role of impurity phases (CuO and Cu) in the photocatalytic process should be investigated;

2) the estimate of the photocatalytic activity should be done relative to some reference (standard) material(s);

3) the optical measurements of the band gap should be done properly or excluded from the manuscript.

Author Response

Response to Reviewer:

The revision is only partially satisfactory. Several important points remain unclear and require further studies as authors mentioned in their reply. Therefore, I cannot recommend the paper for publication in the present form.

We thank the Reviewer for the careful consideration of our work and the provided useful comments.

There are three main problems:

1) the role of impurity phases (CuO and Cu) in the photocatalytic process should be investigated;

Reply: Special thanks to you for your good comments. However, our experimental conditions and time are limited, and we cannot study the role of impurity phases (CuO and Cu) in the photocatalytic process.

2) the estimate of the photocatalytic activity should be done relative to some reference (standard) material(s);

Reply: We kindly thank the Reviewer for the provided idea. The research on the photocatalytic performance of copper nitride films is only a preliminary exploration stage, and it is difficult to compare a standard amount to prove its catalytic activity. Therefore, this article only uses titanium dioxide as a simple comparison.

3) the optical measurements of the band gap should be done properly or excluded from the manuscript.

Reply: We kindly thank the Reviewer for the comment. We believe that the optical band gap value is closely related to its photocatalytic performance, so it cannot be deleted in this manuscript.